Sample-level sound synthesis with recurrent neural networks and conceptors

Kiefer Chris c.kiefer@sussex.ac.uk
Experimental Music Technologies Lab, Department of Music, University of Sussex , Brighton , United Kingdom
Stowell Dan
Electronic publication date: 2019 Jul 8
Publication date: 2019
Volume: 5
Electronic Location ID: e205
Received 2018 Nov 14; Accepted 2019 Jun 13
Copyright: ©2019 Kiefer
Copyright year: 2019
Copyright holder: Kiefer
License: This is an open access article distributed under the terms of the Creative Commons Attribution License, which permits unrestricted use, distribution, reproduction and adaptation in any medium and for any purpose provided that it is properly attributed. For attribution, the original author(s), title, publication source (PeerJ Computer Science) and either DOI or URL of the article must be cited.
License URL: https://creativecommons.org/licenses/by/4.0/

Keywords: Sound synthesis, Machine learning, Reservoir computing, Conceptors, Dynamical systems, Echo state networks

Funding: The author received no funding for this work.

==============================
Conceptors are a recent development in the field of reservoir computing; they can be used to influence the dynamics of recurrent neural networks (RNNs), enabling generation of arbitrary patterns based on training data. Conceptors allow interpolation and extrapolation between patterns, and also provide a system of boolean logic for combining patterns together. Generation and manipulation of arbitrary patterns using conceptors has significant potential as a sound synthesis method for applications in computer music but has yet to be explored. Conceptors are untested with the generation of multi-timbre audio patterns, and little testing has been done on scalability to longer patterns required for audio. A novel method of sound synthesis based on conceptors is introduced. Conceptular Synthesis is based on granular synthesis; sets of conceptors are trained to recall varying patterns from a single RNN, then a runtime mechanism switches between them, generating short patterns which are recombined into a longer sound. The quality of sound resynthesis using this technique is experimentally evaluated. Conceptor models are shown to resynthesise audio with a comparable quality to a close equivalent technique using echo state networks with stored patterns and output feedback. Conceptor models are also shown to excel in their malleability and potential for creative sound manipulation, in comparison to echo state network models which tend to fail when the same manipulations are applied. Examples are given demonstrating creative sonic possibilities, by exploiting conceptor pattern morphing, boolean conceptor logic and manipulation of RNN dynamics. Limitations of conceptor models are revealed with regards to reproduction quality, and pragmatic limitations are also shown, where rises in computation and memory requirements preclude the use of these models for training with longer sound samples. The techniques presented here represent an initial exploration of the sound synthesis potential of conceptors, demonstrating possible creative applications in sound design; future possibilities and research questions are outlined.

Introduction

Machine Learning and Sound Synthesis

Current intersections between sound synthesis and machine learning are evolving quickly. We have seen significant progress in symbolic note generation (e.g., RL Tuner (Jaques et al., 2016), Flow Machines (Ghedini, Pachet & Roy, 2016)), parametric control of sound synthesis models (e.g Wekinator (Fiebrink, 2011), automatic VST programming (Yee-King, Fedden & d’Inverno, 2018)) and also with current state of the art raw audio generation techniques. These recent advances in raw audio synthesis principally use deep architectures, for example WaveNet (Oord et al., 2016), SampleRNN (Mehri et al., 2016), NSynth (Engel et al., 2017), GANSynth (Engel et al., 2019) and WaveGAN (Donahue, McAuley & Puckette, 2018), to generate low-level audio representations (sample or spectral level) without using a synthesis engine, working as self-contained models that merge sound generation and control into one.

There is also significant interest from the computer music community in sound synthesis with dynamical and chaotic systems, with strong connections to RNN techniques being used in contemporary deep architectures. This goes back to the earlier work of composers such as Roland Kayn who composed with electronic cybernetic systems, and is reflected in more recent work from, for example, Sanfilippo & Valle (2013) on feedback systems, Ianigro & Bown (2018) on sound synthesis with continuous-time recurrent neural networks, Wyse (2018) on sound synthesis with RNNs and Mudd (2017) on nonlinear dynamical processes in musical tools.

The work presented here draws on overlapping research in both machine learning and dynamical systems techniques, in the context of sound synthesis.

Reservoir computing

While many contemporary developments in machine learning and sound synthesis are based on deep neural network paradigms, pioneering work has also been taking place within the bio-inspired field of reservoir computing (RC) (Schrauwen, Verstraeten & Van Campenhout, 2007). Within the RC paradigm, computation is performed using a structure that groups an untrained reservoir with a fixed input layer and a trainable output layer. The reservoir is a complex dynamical system which is perturbed by input signals and transforms these signals into a high-dimensional state space, the current state being dependent on both the current input and on a fading history of previous inputs. The output layer performs a linear transformation of the current reservoir state, and can be trained using supervised methods. RC systems can learn nonlinear and temporal mappings between the input and output signals. A reservoir can be created using both physical systems (e.g bacteria (Jones et al., 2007), a bucket of water (Fernando & Sojakka, 2003) or optics (Duport et al., 2016)) and digital systems. The latter usually take the form of liquid-state machines (Maass, Natschläger & Markram, 2002) or echo state networks (ESNs) (Jaeger, 2010).

Echo state networks

ESNs have so far been the primary technique employed for sound and music applications within the RC field. An ESN (see Fig. 1) uses a randomly generated recurrent neural network (RNN) as a reservoir. This reservoir is connected to inputs and output via single layers of weights. The output layer weights can be trained using linear optimisation algorithms such as ridge regression (Lukoševičius, 2012, p. 10).

Figure 1 An example of an Echo State Network with ten sparsely connected nodes, single inputs and outputs, and fully connected input and output layers.

ESNs are inherently suited to audio applications due to their temporal dynamics. Jaeger’s original work with ESNs included examples of models being trained to output discrete-periodic sequences and learning to behave as sine wave oscillators (Jaeger, 2010). Subsequently, ESNs have been applied to a range of creative sound and music tasks. These include symbolic sound generation tasks such as melody generation (Jaeger & Eck, 2006) and generative human-feel drumming (Tidemann & Demiris, 2008); direct audio manipulation and synthesis applications bear examples of amplifier modelling, audio prediction and polyphonic transcription (Holzmann, 2009b; Holzmann, 2009a; Keuninckx, Danckaert & Van der Sande, 2017); they have also been used for modelling complex mappings in interactive music systems (Kiefer, 2014).

Under the classical ESN approach, as applied to the task of sound synthesis, ESNs are trained as audio rate pattern generators. A limitation of the classical ESN approach is that it is challenging to learn multiple attractors, corresponding to the generation of multiple patterns on different timescales with a single reservoir, although Holzmann (2009a) offered a solution by decoupling the reservoir with IIR filter neurons.

A recent development of the ESN paradigm comes in the form of conceptors, an addition to the basic architecture of ESNs that enables the behaviour of the reservoir to be manipulated.

Conceptors

Conceptors (Jaeger, 2014a), offer a highly flexible method for generating and manipulating multiple patterns within single reservoirs. Conceptors are a mechanism for performing a variety of neuro-computational functions, the ones most relevant to sound synthesis being incremental learning and generation of patterns, morphing and extrapolation of patterns, cued pattern recall, and the use of boolean logic to combine patterns (Jaeger, 2014a). They work by learning the subset of state space visited by an RNN when driven by a particular input. They can then be used to restrict the RNN to operate with this subspace, functioning like an attractor (Gast et al., 2017). The separation of an RNN’s state space in this manner allows multiple attractors to be learned using the same network, and for combinations of these subspaces to be used to manipulate the dynamics and output of the RNN. The potential for combination of conceptors is a very powerful feature of this technique, and Jaeger describes boolean logic rules for achieving this (Jaeger, 2014b p.50). Their strong potential for pattern generation, extrapolation and manipulation, and the combination of continuous and discrete-boolean methods of manipulation are compelling reasons to believe they will have strong applications in the field of audio and creative sound production. Sound generation with conceptors has, however, yet to be explored. Jaeger’s original work focuses on generation of short patterns of 20 samples of less, where much longer patterns are required for sound generation. Questions are unanswered concerning whether conceptors will (a) effectively generate longer patterns needed to synthesise audio signals at reasonable sample rates, (b) allow generation and combination of patterns with varied timbres within a single model and (c) produce signals effectively when evaluated with perceptually realistic audio comparison measurements. This paper approaches these questions through the application of conceptor models to a standard sound synthesis technique. It evaluates the effectiveness of conceptors at resynthesising sampled sound, and demonstrates conceptor-based sound synthesis techniques within a granular synthesis paradigm.

New sound synthesis methods

A new method of conceptor-based sound synthesis is demonstrated, named conceptular synthesis. This is a synthesis method based on granular synthesis. Granular synthesis (Roads, 2004) is based on the sequencing, combination and manipulation of short (typically 20 ms–100 ms) windowed segments (grains) of sampled sound. It is a powerful technique for creating and coherently manipulating sound; applications include time and pitch independent stretching of pre-recorded audio. In conceptular synthesis, an RNN model is trained to generate grains, which are recalled by conceptors. The use of conceptor based RNN models allows flexible sound manipulation through creative combinations of conceptors to influence reservoir behaviour. This method is described below; to begin with, a mathematical description of the RNN and conceptor models is presented.

Basic Models

This section summarises the fundamental methods used in the creation of the sound synthesis models described below. For a more detailed explanation of these methods, please refer to Jaeger’s extensive technical report on conceptors (Jaeger, 2014b). The notation used below will be used throughout the paper. Matrices are represented by capital letters, vectors by lower-case letters, and scalar variables are shown using the greek alphabet. x(n) denotes the state of vector x at timestep n.M′ denotes the transposition of matrix M.

The basic model is an RNN consisting of ψ nodes, updated according to Eqs. (1)–(3): (1) zn+1=Wxn+Winan+1

At discrete time step n, activation levels for each RNN node are stored in state vector x of size ψ. The nodes are sparsely connected such that each node is connected, on average, to 10 other nodes (as recommended in Lukoševičius (2012, section 3.2.2). Connection weight values are stored in weight matrix W (of size ψ x ψ). Unconnected nodes share a weight value of 0. An input signal vector a (size 1) is fully connected to the RNN nodes with the input weight matrix Win (size ψ x 1). Win is generated using linear random values between −γinput and γinput. Reservoir weight values are randomly chosen from a normal distribution, scaled according to the spectral radius γW (to limit the maximum absolute eigenvalue), and then optimised during training. In Eq. (1), the activation levels for each node are stored in vector z; in Eq. (2) these activation values are passed through a nonlinearity, and smoothed using leaky integration. (2) xn+1=1−αxn+αtanhzn+1+b

b is a vector of ψ biases, which are generated from a linear random distribution between −γbias and γbias. Scaling in the above cases refers to a tensor being multiplied element-wise by a scalar value, X = Xγ. The tanh smoothing function ensures that the reservoir states remain in the range −1 to 1, and introduces a nonlinearity into each node. α is a leaky integration coefficient (Lukoševičius, 2012, section 3.2.6). This adds a one-pole lowpass filter to each node; lowering α (between 0 and 1) will slow down reservoir dynamics. This parameter can be fine-tuned to align the temporal dynamics of the reservoir to those of the desired output. (3) yn+1=Woutxn+1

Output weights Wout are a matrix of size 1 x ψ, whose values are optimised during training. The output vector y is a vector of size 1.

A model is trained in two phases: (a) audio signals aj are stored (Jaeger, 2014a) in the reservoir, so that they can later be reproduced, (b) a conceptor is calculated for each audio signal. Following training, the model and conceptors are combined and manipulated to synthesise sound.

Storing patterns in the RNN and calculating output weights

In this phase of training, a set of randomly generated reservoir weights W∗ are adapted so that the model can reproduce an array of driving audio signals aj, resulting in a new set of weights W. The number of elements in aj is determined by the sample slicing process detailed below. W∗ is optimised such that (a) Wx(n) ≈ W∗x(n) + Wina(n), i.e., the reservoir can simulate the driving inputs in their absence, and (b) the magnitudes of weights W are minimised. The resulting weight matrix W is used in all further calculations. Win is no longer required after this step.

Training starts with a washout phase of length λ where the reservoir dynamics are allowed to settle, reducing influence from any transients that might result from the initial randomised state and that may adversely affect training. The model is subsequently driven by an input for length ϕ. The size of ϕ is task dependent, but should by large enough to collect the reservoir states that are likely to occur when perturbed by the input sequence. ϕ is calculated as an integer multiple of the length of training signal aj. The training process works as follows: for each pattern aj, the reservoir with weights W∗ is driven from an initial randomised state for λ + ϕ steps using Eqs. (1) and (2), and the resultant reservoir states are collected.

Beginning at timestep 0, the states x from timesteps λ − 1…λ + ϕ − 2 are stored in ψ x ϕ matrix, X ˜j; states x from timesteps λ…λ + ϕ − 1 are stored in ψ x ϕ matrix, Xj; states z from timesteps λ…λ + ϕ − 1 are stored in ψ x ϕ matrix, Mj. The remaining states from the washout phase are discarded. The driving signals aj from timesteps λ…λ + ϕ − 1 are stored in 1 x ϕ matrices Pj.

These collections are concatenated into matrices X ˜=X ˜1|X ˜2|…X ˜n, X = [X1|X2|…Xn], M = [M1|M2|…Mn] and P = [P1|P2|…Pn]

W and Wout can now be calculated using linear regression. The regression could be solved using a number of techniques; in this case, following Jaeger’s initial work on conceptors (Jaeger, 2014b), ridge regression is used: (4) W=X ˜X ˜′+ρWIψxψ−1X ˜M′′

(5) Wout=XX′+ρoutIψxψ−1XP′′

In both of the above, I is an identity matrix and ρW and ρout are regularisation factors.

Calculating conceptors

Conceptors can take several forms, the form used in this study is the alloconceptor (Jaeger, 2017, p18), a matrix conceptor that is calculated after patterns are stored in the network, and inserted into the update loop of the network at runtime. To calculate a conceptor which will influence the RNN to reproduce audio signal aj, the reservoir state correlation matrix R is initially calculated: (6) R=XjXj′ϕ

The singular value decomposition (SVD) of R is found (7) UjSjUj′=Rj

Sj is modified as follows, and used to calculate the conceptor Cj: (8) Snew=SjSj+β−2Iψxψ−1

(9) Cj=UjSnewUj′

β is the aperture of the conceptor (Jaeger, 2014b, p43). The aperture is a scaling factor for the amount of energy that is allowed to pass when the conceptor filters the reservoir state in Eq. (11). The optimal value for β can be found programatically (see below).

The new conceptor can now be inserted to the runtime loop of the RNN, as follows: (10) zn+1=Wxn

This is a modification of Eq. (1), with the audio signal input removed, as it is no longer needed. z(n + 1) is then passed through the leaky integration filter from Eq. (2), and result is multiplied by the conceptor, as follows: (11) x∗n+1=1−αxn+αtanhzn+1+bxn+1=Cjx∗n+1

Following this step, Eq. (3) is used to calculate the output signal.

An optimal value for β can be found by observing the attenuation aC,β, which is the level of reservoir signal energy suppressed by the conceptor when the network is updated using Eqs. (10) and (11) (Jaeger, 2014b, p47). Attenuation can be calculated as follows: (12) aC,β=E||zn−xn||2E||zn||2

The optimal value for β corresponds to the minimum value of attenuation, calculated by collecting states from the model using conceptors calculated with varying values of β.

The result of this training process is an RNN coupled with a set of conceptors; this model is referred to as a conceptor-controlled recurrent neural network (CCRNN).

These basic methods are used in the experiments below, and expanded on with new techniques that allow training and exploitation of the models for sounds synthesis.

Method and Materials

This project asks how the pattern generation ability of CCRNNs can be applied to the field of sound synthesis. It aims to establish and evaluate the fundamental capabilities of CCRNNs to be trained to reproduce arbitrary audio signals, and to explore their creative affordances. The next section evaluates the potential of CCRRNs to resynthesise sampled sounds.

The ability of trained models to resynthesise the training signal is used as a measure of basic success in sound synthesis. Resynthesis ability is the core indication of sound synthesis quality, although this evaluation only tells part of the story, as the techniques outlined in this project are intended for open-ended use in creative sound synthesis applications. To this end, the project maps out key methods for parameterising and manipulating CCRNN sound synthesis models to create new sonic variations of the original training material, establishing the technical strengths and limitations of CCRNN sound synthesis, and identify open questions for future research in this area.

There is some discussion of processing time to indicate the scale of computation involved with these techniques. The conceptular synthesis experiment was run on a machine with 4.2 GHz i7 CPU and an NVidia GTX 1080 Ti GPU using TensorFlow.

Python 3 source code in Jupyter notebooks for all experiments is provided at https://github.com/chriskiefer/conceptorSoundSynthesis. Source code for a working implementation of conceptular synthesis in the form of a drum synthesiser can be found at https://github.com/chriskiefer/conceptularBeatSynth.

Error and similarity metrics

The experiments in this project focus on the quality of audio signals resynthesised with trained models in comparison to the original training material. In wider literature in reservoir computing, Normalised Root-Mean-Square Error [30 p2] (NRMSE) is commonly used to measure similarity between signals. NRMSE does not reflect perceptual aspects of sound similarity; these are crucial to understanding the results therefore a different metric is required. Mel-Frequency Cepstral Coefficients (MFCCS) have been shown to be a robust measure of timbral similarity (Jensen et al., 2009), and a good model of perceptual timbre space (Pampalk, Dixon & Widmer, 2003). They are widely used for music information retrieval tasks (Hamel & Eck, 2010) across a variety of use cases (e.g., (Yee-King, Fedden & d’Inverno, 2018; Khunarsal, Lursinsap & Raicharoen, 2013).

For the purpose of audio signal comparison, MFCCs were calculated from a 2048 point windowed FFT. The sounds being analysed were short (typically between 200 and 5000 samples); in order to capture the detail of the timbral envelope at all stages, a short 64 sample hop-size was used. The FFT results from each window were used to calculate 20 MFCCs. The first MFCC coefficient from each window was discarded as it does not give information about timbre (Jensen et al., 2009). The remaining coefficients from each window were appended to create a single feature vector. The feature vectors from two sounds were compared using NRMSE to arrive at an error value that reflects the similarity between the two sounds. This will be referred to as the MFCC error, with lower values indicating higher similarity.

Where relevant, waveforms and spectrograms are displayed for visual comparison, and audio is included in the dataset accompanying this paper.

Conceptular Synthesis

Conceptular synthesis is a new sound synthesis method based on CCRNNs, and expanding on an established method of sound synthesis, granular synthesis. In granular synthesis, sound is broken into small parts (grains), which are recombined in varying ways to produce new sounds. The theoretical roots of this method lie in Gabor’s (1947) theory of acoustic quanta, and in the compositional theory of Xenakis (1971). Digital implementations of the technique were developed by Roads (1978) and Truax (1986). Granular synthesis offers methods for further sound manipulation techniques including timestretching (Truax, 1994) and corpus-based concatenative synthesis (Schwarz, 2006).

Jaeger’s demonstration of the ability of CCRNNs to be trained to generate arbitrary sequences suggests that they could become powerful sound synthesis tools, as they can theoretically reproduce arbitrary audio waveforms. However they are pragmatically limited to playing relatively short sequences; the reason for this is that the computational complexity of the model increases with ψ, and the desired size of ψ increases with the length of training sequences. However, if a model is trained to reproduce a set of shorter sound sequences, then granular synthesis techniques can be used to recombine these sequences to produce longer sounds. Conceptular synthesis therefore expands upon granular synthesis, by dynamically generating grains using conceptors rather than replaying grains from sound sample data. Grain patterns are stored in an RNN, and conceptors force the RNN to replay specific grains. A granular synthesis-style control mechanism is used to switch conceptors so that the model generates a sequence of short signals, which are combined into a longer waveform. The use of dynamic models instead of static sample data broadens the sonic potential of conceptular synthesis in comparison to classical granular synthesis, as the models provide further possibilities for creative manipulation.

This section begins by describing conceptular synthesis techniques in detail, illustrated by an example demonstrating resynthesis of a kick drum sample. Resynthesis quality is then explored in an experiment comparing conceptular synthesis to a baseline method using feedback induced oscillation. Following this, extended sound synthesis options are described.

Conceptular synthesis techniques

Conceptular synthesis works by subdividing the audio training data into a set of sub-sequences, and learning an RNN and set of conceptors that can regenerate these sub-sequences, with the intention of resynthesising the audio sample by recombining the model-generated sequences. The audio training data may be a single audio sample, or it may be composed of multiple audio samples to create a model that will create variation between different samples.

Slicing training data

Two methods of slicing the audio were explored: using constant or variable values for the sub-sequence size μ. Using constant μ, the sample is divided into a set of equal length signals a, of length μ samples each. The optimal value of μ for a particular sound sample can be determined programmatically through a grid search. The constant μ slicing method runs into problems, particularly when there are dominant frequencies in the source audio whose wavelength is longer than μ. Slicing lower frequency waveforms at non-zero points creates high-frequency artefacts in the training data, which can distort the training process because these artefacts are not present in the original training material. For example, consider the kick drum waveform in Fig. 2. The sample has low frequency components with varying long wavelengths, and there is no constant value of μ that will avoid slicing at non-zero points. To avoid this, the sample can be analysed using a zero-crossing detector, resulting in a set of points i at which the sample is sliced to create a set of driving audio signals a. (13) i=n|yn⩾0∧yn+1<0.

Hyperparameters and training

A key parameter is the reservoir size ψ.ψ correlates with the memory capacity of the reservoir; it should be at least equal to the number of independent variables needed for the task the model is being trained for (Jaeger, 2002). As we increase the quantity and length of training signals, we need to increase ψ to give the model the capacity to learn them. However increasing ψ makes computation more expensive (at approximately O(N2)), so there are practical limits to this value.

The other key parameter is the leak rate α. Lowering α has the effect of filtering out high frequencies in the reservoir activation levels x, and therefore slowing down the behaviour of the RNN. α needs to be chosen such that the model can reproduce the frequency content from the source audio, for example a sound with dominant low frequencies like the kick drum in Fig. 2 will need a network with slowly changing activations, and therefore a low α value. Optimal values can be found using a grid search.

After choosing hyperparameters, a model is trained using the techniques set out earlier in the paper. The output is a set of conceptors, calculated to reproduce each audio signal aj.

Resynthesis

To resynthesise the training signal, the CCRNN is initially run for λ steps with the first conceptor C0. The model is then run with each conceptor Cj inserted into the update loop, as described in Eqs. (10) and (11), to create a set of output signals q. The number of samples for which each conceptor is used in the runtime loop corresponds with the length of the audio signal from which the conceptor was trained. The algorithm makes a short linear crossfade between conceptors, over a small percentage of the pattern length. This prevents artefacts appearing from instantaneous switching of conceptors. Finally, output signals are appended to create the waveform k = [q0|q1|…|qn].

Example: resynthesis of kick drum

The output of a trained conceptular synthesis model is now shown, to illustrate how this technique can be applied. A CCRNN model was trained to resynthesise a kick drum. The original audio (see Fig. 2 and Audio S5) was re-sampled at 22050 Hz (half of CD-quality), in order to reduce the CPU load of training.

The sample was segmented using the zero-crossing method, and the model was trained with the parameters as shown in Table 1, with leak rate α = 0.15 . The kick drum sample was resynthesised with the crossfade length set at 5% of signal length. The result is shown in Figs. 2 and 3, and included in Audio S7. Both show a close reconstruction of the original, with the addition of some high frequency artefacts.

Figure 2 A comparison of the original kick drum sample and the output of the trained CCRNN model.

Table 1 Hyperparameters in the resynthesis quality experiment.

Model Type	ψ	α	γW	γinput	γbias	λ	ϱW	ϱout	
CCRNN	900	{0.05, 0.15...0.95}	1.5	1.2	0.3	50	1e − 5	1e − 5	
ESNSPF	20...900	0...1	1.0	0...1	0...1	50	1e − 5	1e − 5	

Figure 3 Spectrograms comparing the resynthesised kick drum (A) to the original sample (B).

This example is not compared quantitatively to the original, instead this is done systematically in the experiment below.

Measuring resynthesis quality

We have seen the potential capabilities for conceptular synthesis to resynthesise audio from trained CCRNN models. These trained models offer extensive creative possibilities for sound synthesis, which are detailed later in the paper. Before exploring these avenues, the quality of resynthesis method should be evaluated, in comparison to an existing baseline.

Baseline models: echo state networks with stored patterns and feedback

The closest comparable method to conceptular synthesis is to use ESNs with output feedback as trainable oscillators. Feedback is used to guide reservoir dynamics to a stable limit cycle where the model is outputing a desired signal. Early research on ESNs (Jaeger, 2010) showed their capability for self-driven oscillation using output feedback. More recent research has focused on increasing the effectiveness of output feedback by adapting the weight matrix W to increase the accuracy of reproduced patterns. Jaeger (2014a) p2 summarises this family of techniques including self-prediction (Mayer & Browne, 2004), self-sensing networks (Sussillo & Abbott, 2012) and Jaeger’s own method for storing patterns which has already been detailed in this paper. The reservoirs for both techniques are created using the same stored pattern method, but then the methods diverge. These baseline models will be referred to as ESNSPFs.

The training method for ESNSPFs works as follows: a set of audio signals is stored in an RNN, using an identical approach as detailed earlier with Eqs. (1), (2) and (4). The result of this process is to adapt the randomly initialised weight matrix W∗ to trained matrix W.

This model is now used for training a set of output weight matrices, such that each matrix Woutj will be used to recreate audio signal aj. To train an ESNSPF using output feedback, we train an output layer that will predict the the next sample from the input signal. The model is driven with signal aj using Eqs. (1) and (2). Following a washout period, states x(n) from timesteps λ to λ + ϕ − 2 to are stored in ψxϕ − 1 matrix X. Samples from the audio signal aj from λ + 1 to λ + ϕ − 1 are stored in 1 x ϕ − 1 matrix P. The output weight matrix Woutj can then be calculated using Eq. (5). Feedback models are sensitive to initial conditions; a brute force search can be used to find a good value for the initial state x(0), stored to use later as xcue.

At runtime, the model is run by feeding the output back into the input with a modification of Eq. (1), starting from initial state xcue. To reproduce audio signal aj, the model is updated as follows: (14) zn+1=Wxn+Winynxn+1=1−αxn+αtanhzn+1+byn+1=Woutjxn+1

To recreate a sequence of patterns or grains that comprise a longer audio signal, each output matrix Woutj is sequentially inserted to the runtime loop for a period matching the length of the audio signal aj.

Dataset

Sounds from the Ixi Lang data set (Magnusson & Kiefer, 2019) were used to compare the two methods. This is a collection of the sounds that accompanies the Ixi Lang live coding environment. The collection represents a wide variety of short samples that are used for live performance. There are 127 audio clips, lasting between 5ms and 36.3s (mean: 1.75s).

Method

Each sample was resampled at 22,050 Hz, normalised and scaled by half, and truncated to a maximum of 5,000 samples (or 0.23s); this was to create a balance between computation demands on the model, and providing enough material to make a useful comparison. The sample was then sliced; the zero-crossing method (Eq. (13)) was used as this is more widely applicable to a range of timbres. A maximum of 150 patterns were kept from the slicing process, again to reduce demands on computation time and memory for storing conceptors. This process resulted in a set of patterns extracted from each sample in the dataset. ESNSPF and CCRNN models were trained to resynthesise each pattern set, and then used to resynthesise the corresponding samples.

Table 1 shows the hyperparameters used for both models types, chosen to match the models as fairly as possible for comparison. Some were fixed, others were searched for within the ranges shown. Fixed hyperparameters were chosen based on experimental reports in wider literature (Lukoševičius, 2012; Jaeger, 2005; Jaeger, 2014b), and through extensive manual experimentation. Some parameters were deemed to be sensitive to the training material, for example α needs to be tuned to match the frequencies in the source, in which case the values were optimised through automatic search as detailed below.

The use of randomly determined weight values is fundamental to the design of both models, however this leads to variance in training results. To mitigate against this variance, the training process was run multiple times at each hyperparameter setting and the best models chosen. It is acknowledged that, as with all non-deterministically generated models, it is possible that the most optimal model will not be found, but running the training process multiple times will increase the possibility of finding a better model. Further to this, variances in this process will be averaged out across the 127 samples in the dataset.

CCRNN Models The key hyperparameter for this model type was the leak rate α, which needed to be tuned to match the frequency response of the model with the frequencies needed for resynthesis of the training material. For each sample in the dataset, α was determined through a grid search (as recommended in (Lukoševičius, 2012, section 3.3.4)) of values {0.05, 0.15, 0.25…0.95}. The grid search evaluated models at each of these settings at a lower model size ( ψ = 600) in order to save computation time, evaluating 5 different models and recording the best MFCC error. This process resulted in an optimal value of α which was used to evaluate 10 models at a higher model size ( ψ = 900). The highest score of these models was recorded.

ESNSPF Models Models trained with output feedback showed sensitivity to four parameters: model size ψ, leak rate α, bias scaling γbias and input scaling γinput. Interestingly, this technique showed sensitivity to ψ, while CCRNN models show consistent improvement with larger ψ. A four-dimensional grid search was impractical due to computational demands, instead a microbial genetic algorithm (Harvey, 2009) was used to find optimal values, conducting a stochastic evolutionary search of parameters in the ranges shown in Table 1. The fitness function selected the best model based on evaluations of 10 models created with identical hyperparameters. Each model was then evaluated with 10 randomised starting states xcue, and the xcue with the lowest error was chosen.

Results

The experiment resulted in an error score for each model type for each of the 127 samples in the dataset. The two methods gave comparable results (Fig. 4) (CCRNN: mean 0.542, median 0.482, ESNSPF: mean 0.568, median 0.536). There was no significant difference between the model types (wilcoxon signed rank test: W = 3872, p=0 .644). These results are discussed further below, after contextualisation within extended synthesis techniques.

Figure 4 A violin plot of MFCC error scores for each model type, for resynthesis of samples in the Ixi Lang dataset.

The plot shows the distribution of scores, with real values marked by vertical lines.

Extended conceptular synthesis techniques

The above experiment demonstrates that CCRNN models are capable of sound synthesis quality comparable to ESNSPFs. Moving beyond this baseline, they offer a wider range of creative sound synthesis possibilities.

Extended sound synthesis parameters

The sound generation algorithm can be manipulated with three key parameters: speed, leak rate scale, and weight scaling. The speed parameter changes the amount of time in which the algorithm waits until a new conceptor is plugged in to the RNN update loop. For example, a speed of 0.5 results in two cycles of a pattern being played for each conceptor Cj and resulting in a rendered sample that is twice the length of the original. At negative speeds, a sample can be crudely reversed by playing the patterns in reverse sequence. This parameter can have the effect of timestretching, i.e., extending or compressing the length of a sound, independent from its pitch. Audio S8 presents an example of timestretching from 50% up to 800% in 50% steps with a CCRNN model trained on a kick drum sample.

The leak rate α can be scaled during resynthesis, by updating Eq. (2) as follows: (15) αscaled=αξαxn+1=1−αscaledxtargetn+αscaledtanhxtargetn+1+b

ξα becomes an useful parameter in resynthesis for control over timbre and pitch (which is explained later when discussing pitch-controlled oscillators). It should be limited such that α stays between 0 and 1.

Weight scaling can be introduced by updating Eq. (1) so that the weight matrix is linearly scaled by scalar ξW: (16) Wscaled=WξWzn+1=Wscaledxn+Winan+1

This causes timbral changes in the rendered sample whose characteristics are based on the random make up of the RNN. There is some consistency in this parameter in that when raised, more high frequency content tends to be introduced. At higher values, the RNN can behave in musically interesting non-linear ways. Below a lower limit (model dependent), the model output tends towards silence.

Further manipulations of sound can be achieved by manipulating conceptors.

Extending sound synthesis with conceptor logic

The use of conceptor logic and conceptor manipulation is where this mode of sound synthesis significantly moves on from standard granular synthesis features, and brings its own unique possibilities. New conceptors can be created using boolean logic rules; Jaeger (2014b, p.52) defines formulae for AND, OR and NOT operations. Boolean operations provide a system of logic with which to combine conceptors, with applications in classification and memory management. In the case of conceptular synthesis, logic operations provide a wide range of creative possibilities. Conceptors can be logically recombined to create new timbral variations. Two examples are now given:

Example 1

A CCRNN model was trained to reproduce a snare sample (Audio S1). Each conceptor in the snare model Cj was combined with the subsequent three conceptors to make a new set C2, using the rule C2j=Cj∨Cj+1∨Cj+2∨Cj+3 This resulted in a variant on the original snare sound shown in Fig. 5 (Audio S9).

Figure 5 The waveform of a variant of a snare sample, produced using boolean logic C2j=Cj∨Cj+1∨Cj+2∨Cj+3.

Example 2

A new set of conceptors C3 was made by combining each conceptor in the set with a random choice of two other conceptors in the set C3j=Cj∨Crandom1∨Crandom2. This is designed with the intention of keeping the main structure of the sample but introducing random variations. Fig. 6 shows the resulting waveforms from 4 separate iterations of this process using the snare model detailed above (Audio S10).

Figure 6 Waveforms showing generative variants of the snare sample, using the boolean logic rule C3j=Cj∨Crandom1∨Crandom2.

In both these examples, the variations are subtle, and the renderings suffer from some audible artefacts, however this does point to generative possibilities that are worthy of further research.

Sound morphing with interpolated conceptors

Jaeger (2014b) p.42 demonstrated shape morphing between heterogeneous patterns using conceptors. This same technique can be applied within conceptular synthesis to morph between sounds. It is not within the scope of this paper to analyse the quality of sound morphing using conceptular synthesis in comparison to other techniques, rather just to demonstrate that sound morphing is a creative possibility with CCRNNs, and to outline how it can be achieved.

Morphing can be implemented by creating a linear combination of conceptors to interpolate between the two patterns the conceptors were trained to recreate. Equation (17) shows how this can be done with two conceptors, where ω is the morphing factor. (17) xn+1=1−ωCi+ωCjtanhWxn+b

Varying ω between 0 and 1 forces the RNN to create a morph between the patterns represented by the two conceptors. When 0 ≤ ω ≤1, the mix of conceptors will interpolate between patterns. However, when ω is outside of this range, the mix of conceptors can extrapolate between patterns.

The intention of morphing between sounds is to create a new mixture of sounds that retains the shared perceptual properties of the original sources (Slaney, Covell & Lassiter, 1996). Morphing was investigated with conceptular synthesis by training a CCRNN recreate patterns from two different samples: the snare sample already discussed, and a short bongo sample (Audio S12). An 800-node network was trained for recreation of 100 individual patterns (fixed length 15) for each sample, resulting in two sets of conceptors, Csnare and Cbongo. Morphing was achieved by creating a new set of conceptors based on a linear mixture of the trained conceptors, for each pattern segment. The results demonstrate a morph between samples that is different from a linear mixture of the two samples. Figure S1 shows how the time-domain waveform result varies over an 11 point morph from ω = 0 to ω = 1, and the result can be heard in Audio S3. For comparison, Fig. S2 and Audio S4 demonstrate a linear mix between amplitude values with the same two samples.

Boolean conceptor logic can also be used for sound morphing. For example, a set of conceptors CbongoSnare was created, with each element combining elements from the snare and the bongo CbongoSnarej=Cbongoj∨Csnarej. A sample rendered with this conceptor set contains characteristics of both sounds (Audio S11).

Equivalent techniques with ESNSPF models

Where possible, equivalent extended sound manipulation techniques were attempted with ESNSPF models. There is no parallel using ESNSPF models for conceptor logic. However, it is possible to manipulate ξα and ξW during resynthesis, and also to introduce the same timestretching mechanism as for CCRNN models. Attempts at all of these methods however did not lead to satisfactory results. Timestretching with ESNSPFs only works when slowing down playback at integer subdivisions of the original speed. At other ratios, the models tend to quickly converge to silence. When altering ξα and ξW, the models tend towards producing high-amplitude artefacts. To demonstrate this, all models trained in the experiment above were tested with either ξα or ξW set to values {0.7, 0.75…1.3}. Figure 7 shows the average percentage change in standard deviation of amplitudes of resynthesised waveforms compared to a waveforms generated with ξα = 1 and ξW = 1. For ESNSPFs, this change in standard deviation is high compared to the same measurement for conceptular synthesis models, reflecting high amplitude artefacts introduced by the change in these values. CCRNN models however remain close to the original amplitude range (although with some very small variation).

Figure 7 Demonstration of the effects of ξα and ξW on waveform amplitude, reflecting high-amplitude artefacts in the ESNSPF models.

Discussion

This work demonstrates how CCRNNs can be used to resynthesise short samples by dividing the sample up into short signals and training a conceptor for the reproduction of each one. An experiment on resynthesising the Ixi Lang dataset show that conceptular synthesis can achieve comparable resynthesis quality to echo state network models that use stored patterns and feedback, when measured using MFCCs. Both types of model could not perfectly reproduce the training samples, but were able to make reasonable reconstructions. There was some variance in resynthesis quality across the results, the causes of which are the topic of future investigation.

CCRNN models offer malleable sound synthesis possibilities when manipulated using inherent runtime parameters and through conceptor combinations created either by interpolation or by boolean logic. These techniques provide unsatisfactory results when attempted with equivalent implementations in ESNSPF models. This is likely to be due to the high sensitivity to initial conditions of models that use output feedback. While they can provide good quality resynthesis, ESNSPF models are brittle in nature, while CCRNNs show themselves to be highly robust to manipulation during resynthesis, making them extremely valuable as creative tools.

There is natural variability between models, due to random initialisation of the RNN. This variability is minimised when using the network within normal constraints, however when pushed into non-linear modes of behaviour by, for example, changing the value of ξW, a higher variability between different CCRNNs can be observed. This behaviour for a particular network may turn out to be musically interesting, lending conceptular synthesis potential for serendipitous discovery of new sounds, and a level of generative unpredictability that is often valued by musicians (McCormack et al., 2009).

Is it possible that, by using the reconstruction error as the evaluation metric, this experiment has produced models that are flawed because of overfitting? In this context, overfitting would be the production of generative models that are only useful for reproducing the training data, but do not function well as malleable creative tools when manipulated using the techniques described above, ie. they would have a limited aesthetic state space (Eldridge, 2015). Conventionally in ESN research, the complexity of reservoirs, approximately determined by the size of N, has been established as a factor in overfitting in discriminative models (Wyffels & Schrauwen, 2010); however, there is very little research on the nature of overfitting in generative ESNs and related models, especially with regards to creating malleable models. Jaeger acknowledges that there are open questions around the concept of overfitting (Jaeger, 2005). The experimental results do not indicate that N is a factor in overfitting in this experiment. In the search for ESNSPF models, N was optimised through evolutionary search. The resulting values of N follow an approximately flat distribution across the search range (mean 379, std: 256). If higher N resulted in higher scoring (but overfitted) models, these values would have been skewed towards larger model sizes. Further investigation revealed no significant correlation between N and reconstruction error. The measurement of high amplitude artefacts produced by manipulation of ξα and ξW (as detailed above) could be taken as a metric for basic malleability. There was no significant correlation between N and the level of artefacts, overall showing a lack of evidence for a connection between N and the possible effects of overfitting in ESNSPFs. CCRNN models were produced with a fixed N, and have shown themselves to be highly malleable in the examples detailed in this project. When producing generative reservoir models, increasing N could give the model more opportunity for varied dynamical behaviours rather than limiting scope; it’s certain that overfitting in generative reservoir models is a nebulous concept that warrants future attention.

An audio synthesis process would ideally run in realtime. In this example, the rendering was carried out on a CPU, and was around 10 times slower than realtime. While this leaves much room for improvement, it should be noted that this version was not optimised for speed, and a dedicated C++ or GPU renderer is expected to be faster than the python version used here. It does however show the scale of computation involved in this method of sound synthesis, and indicates that computational resources are a challenge in this area.

Conclusions and Future Work

The experiments presented here show how recurrent neural networks under conceptor control, as originally described in (Jaeger, 2014b), can be configured, trained and run as sample-level sound synthesisers. Conceptular synthesis is an extension of granular synthesis, where a CCRNN is trained to reproduce very short segments of a sound sample using conceptors to recall the different patterns. It is controlled at runtime to recombine these short segments into a longer continuous sound. The models were not limited to straight-forward sound reproduction; CCRNNs presented a large variety of creative options for synthesising new sounds based on the training materials. Techniques included classic granular synthesis methods: timestretching and compression, and creative recombination of grains. Techniques were extended by the new possibilities of combining conceptors, using boolean conceptor logic, and using linear combinations of conceptors to morph between signals. The leak rate of RNN nodes and the RNN spectral radius can be manipulated at runtime to create new sonic possibilities.

CCRNNs were shown to have similar resynthesis quality to baseline ESNSPF models, when compared using MFCCs. CCRNNs however excelled in their possibilities for creative sound manipulation, compared to ESNSPFs which produced either significant artefacts or silence when manipulated.

The experiments outlined common limitations of CCRNN models for sound synthesis. There was always some high-frequency loss in the reproduction of the original driving audio signals, some further experimentation is needed to discover the source of this issue. Issues with high frequencies are affected by the choice of leak rate α, which needs to be chosen carefully to slow down RNN dynamics for reproduction of low frequency patterns, while also preserving enough high-frequency dynamics. It’s possible that oversampling may help, although the efficiency impact of oversampling could be significant considering the high computational cost of training and running these models.

This work helps to answer the questions posed at the beginning of the paper concerning the fundamental capability of conceptors to synthesise audio signals. Conceptors are able to generate longer patterns needed for audio signals at reasonable sample rates, as demonstrated by resynthesis of simple sine-like patterns up to 890 samples long in the example of the kick drum, and resynthesis of varied audio materials in the Ixi Lang dataset. These pattern lengths are relatively short but useful enough for sound synthesis. As pattern lengths extend, pragmatic limits on computational resources limit further exploration. The resynthesis quality experiment established that when training models with multi-timbre sounds, there is variance in the ability of CCRNN models to reproduce these sounds accurately, indicating sensitivity in CCRNNs to sonic qualities in the source materials. It’s not clear yet what this relationship is; this should be the topic of further investigations.

Conceptular synthesis required large models (of approximately 800 nodes upwards) to produce reasonable results, resulting in slow resynthesis times. The large size of these models was required for them to be able to learn either long patterns or high volumes or short patterns. The technique ran at around 10 times slower than realtime. The memory requirements for conceptular synthesis were particularly large, as a conceptor was needed to reproduce each training signal, resulting in model sizes between 0.5 and 1 GB in the experiments above. These computation requirements still may be considered lightweight compared to some deep learning sound-synthesis techniques, nevertheless it would be a considerable success if these models could be optimised to reach realtime at reasonable sample rates. Recent research into deep architectures in echo state networks may offer promise for increasing computational efficiency, as they have been shown to have better memory capacity compared to classical ESNs with similar numbers of nodes (Gallicchio, Micheli & Silvestri, 2018). More broadly, the relationship between memory capacity and computation time will be a limit on sound synthesis with CCRNNS and their potential to move beyond short sound samples, until methods are found to change architectures and reduce this dependency.

This initial demonstration of the potential of sound synthesis with CCRNNs stimulates further questions. Future research should establish:

1. how these techniques can be scaled upwards to facilitate learning models of longer sound samples;

2. the causes of variance in resynthesis quality when training models with signals of varied timbre;

3. whether high-frequency loss in resynthesis can be resolved;

4. how to optimise the RNN leak rate α for sounds with wide frequency ranges;

5. how the techniques identified in this paper can be extended for the purpose of generative sound synthesis;

6. how to optimise network architectures to achieve closer-to-realtime performance;

7. how to conceptualise overfitting, in the context of producing creatively malleable generative models;

8. potential for use with analysis and resynthesis in the spectral domain (as we are seeing with systems such as NSynth (Engel et al., 2017), GANSynth (Engel et al., 2019)).

Conceptually, CCRNN architectures could be creatively compelling for computer musicians; it can sometimes be challenging to create believable and coherent complexity with standard digital sound generation and editing tools. With CCRNNs, complexity comes for free and needs to be managed instead of created. The models presented here are inherently variable, and can be easily encouraged towards unpredictability and nonlinearity, creating sometimes surprising and serendipitous results. The models offer plenty of entry points for creative manipulation, with a potentially wide aesthetic state space (Eldridge, 2015). The musician must interact with these models, rather than control them.

The experiments presented here have mapped out initial explorations into sound synthesis with CCRNNs. They extend a classical sound synthesis method, bringing boolean logic, pattern morphing and non-linear modulation possibilities into granular-style synthesis. The techniques exhibit some limitations that require further investigation, but also show unique creative possibilities for musicians, and rich potential for further research in this area.

Supplemental Information

Audio S1 A snare sample, used in examples of extended synthesis technniques

Click here for additional data file.

Audio S2 A 65Hz square wave oscillator, resynthesised by a CCRNN model

Click here for additional data file.

Audio S3 An 11-point morph between snare and bongo samples, using interpolated conceptors

Click here for additional data file.

Audio S4 An 11-point time-domain mix between snare and bongo samples

Click here for additional data file.

Audio S5 A kick drum sample, used in the example of resynthesis with CCRNN models

Click here for additional data file.

Audio S6 Resynthesis of a square wave with a CCRNN model, as alpha is raised linearly from 0 to 2

Click here for additional data file.

Audio S7 A kick drum sample, resynthesised by a CCRNN model

Click here for additional data file.

Audio S8 An example of timestretching with a CCRNN model; a kick drum is stretched from 50% to 800% of its original length in 50% steps

Click here for additional data file.

Audio S9 The waveform of a variant of the a snare sample, produced using boolean logic

Click here for additional data file.

Audio S10 Four generative variants of a snare sample, created using the boolean logic for combination of conceptors

Click here for additional data file.

Audio S11 A morph between a bongo and a snare, created using conceptor logic

Click here for additional data file.

Audio S12 A bongo sample, used in examples of sound morphing

Click here for additional data file.

Figure S1 Waveforms showing a morph between snare and bongo samples, using interpolated conceptors

The waveforms represent each stage of an 11-point morphing process.

Click here for additional data file.

Figure S2 Waveforms showing a linear amplitude mix between snare and bongo samples

The waveforms represent each stage of an 11-point mix of amplitude values.

Click here for additional data file.

Thank you to Sussex Humanities Lab for generous access of their computing facilities.

Additional Information and Declarations

Competing Interests

Author Contributions

Data Availability

The author declares there are no competing interests.

Chris Kiefer conceived and designed the experiments, performed the experiments, analyzed the data, contributed reagents/materials/analysis tools, prepared figures and/or tables, performed the computation work, authored or reviewed drafts of the paper, approved the final draft.

The following information was supplied regarding data availability:

Code is available at GitHub:

https://github.com/chriskiefer/conceptorSoundSynthesis

https://github.com/chriskiefer/conceptularBeatSynth.

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
