# Peer review of "Sample-level sound synthesis with recurrent neural networks and conceptors"

_PeerJ Computer Science, doi:10.7717/peerj-cs.205_

## Round 0.1 · original submission · Major Revisions

The reviewers felt the content was interesting and novel.

All three reviewers commented that the paper needs to be tightened up: both in general readability, and in clearly framing the research objectives. As well as the specific points of feedback from the reviewers, please do try to do this.

Two reviewers also commented on the limited evaluation. For resynthesis experiments (if not the novel synthesis experiments) it should be straightforward to include baselines and/or to use alternative figures of merit - the reviewers give specific suggestions for both of these, which seem well-motivated.

Reviewer 1 ·

Basic reporting

Overall, the English use is good. The main problem is the readability of the paper. It is very lengthy and more like an experimental report. The paper needs to be reorganized and needs to be more concise.

Experimental design

I have major concerns about the metrics used in the paper, as well as about the lack of baselines. Please see my comments below.

Validity of the findings

I think the applications can be described more clearly.

Additional comments

In the Manuscript, the Author studies generative audio modeling using recurrent neural networks under conceptor control. I think the material is interesting and novel, but there are various fundamental missing points for the paper to be accepted. Below are my detailed comments:

- The Abstract is very vague and general. Please state contributions more clearly and quantitatively, e.g. emphasizing on some of the results on synthesis accuracy.
- I couldn't understand what imposes sparsity in the weight matrix W, please explain.
- Please formulate the operation of scaling with the described factors precisely.
- Please describe the washout superscript in L.
- It is common to solve regression using gradient-descent based methods to avoid the high complexity of matrix inversion. I suggest the Author to consider and discuss this aspect.
- Please define the notations for linear algebra operations like transpose explicitly.
- For sound, mean-squared error on the raw time domain waveform has a very low correlation with the perceptual quality. For example, phase errors causing a shift in the waveform can significantly increase this error, but does not change the perceptual quality much. I suggest the Author to consider this point and evaulate the synthesis quality with other metrics that are commonly used in the literature, like the spectral errors defined on spectrograms.
- There are no baselines for the results, so the efficacy of the proposed RNN architecture is not convincing. For example, can a standard RNN without conceptors achieve a similar quality?
- The results section is very lengthy, both in terms of figures and text. For example, what is the significance of Fig. 3? I suggest the Author to reorganize the section in a more concise way.
- Sound morphing is described but the quality of the morphed waveforms is not evaluated. So it is not convinving whether the proposed technique can synthesize realistic artificial sounds.

·

Basic reporting

The basic reporting meets the standards for PeerJ.

This paper presents two related sound synthesis methods based on "conceptors", a mechanism for learning and then manipulating subsets of the reservoir used in RNNs to create different attractors, and this different pattern generators. The author calls the two methods are conceptular synthesis, and "the conceptillator"

This is an interesting direction of research since recent and ongoing developments in neural networks naturally hold great promise for contributing to sound synthesis. The references and landmarks cited to orient the current work are thorough and appropriate as is the background discussion of Reservoir Computing and Echo State networks.

The organization needs some work, and there are a few issues that need to be addressed to improve clarity (see General Comments to the Author).

Experimental design

The 5 experiments reported are in line with the objectives of the paper. The goals and scope are clear, and the work is carried out well, source code is provided, and (very importantly) sound examples are provided.

Validity of the findings

I think the paper is pretty forthcoming about both the potential and limitations of the synthesis technique. I really look forward to hearing some morphs between complex sounds (even just the "oo" from the spoken "two" and square wave would be interesting).

Additional comments

The paper could be improved for clarity, although some of the questions that arise in the mind of the reader early on in the paper do get clarified later and on subsequent readings, but it would be possible to significantly increase clarity for the first time reader by moving some of the explanation of terms and the discussion of the how the system is used to generate sound up towards the front of the paper.

The clarity in the equation section, "Basic Models" could be improved:

Why the extra text of parens in both parts of the sum in equation 2?

The word "target" in eqs 1-3 is unexplained. I guess it just means the value of the state nodes prior to their alpha decay, but why should your reader guess? Also, "target" is so commonly used to signify a desired output for training, that it is a little confusing here if left unexplained.

123: The nodes are sparsely connected with a probability of 10.0/N in weight matrixW (of size N x N).
Why 10/N? It's a strange and arbitrary number which also doesn't really make sense as N gets small anyway. And how big is N typically? Is it chosen arbitrary? Is this what you later refer to as "the number of nodes in the RNN"?

135, 138: "loaded in to the model" - does this mean "used to train the model"?

144: What is L? What does "washout" mean? How big is L^(train) typically? What is the significance of different values of this parameter?

158: eq(6) is L superscript(train) different from L subscript(train)?

160: S^j is "modified"
Why? Intuition?

161: \alpha is the aperture of the conceptor
Is this different from the \alpha used for the leaky integration from line 131 and eq(2)? If so, better use different notation.
Can you give some words to help with the understanding of the term "aperture" here?

eq(12) - are C and alpha both subscript on 'a' here? So for any C (based on the SVD) and aperture \alpha, you get an 'a', a minimum value of attention which is used ..... how and where?

192: [The percussion sequencer website] On my fast university connection, it too about 12 minutes for the web page to load. Then with the default 32 steps, and 36ms step length, I hit render and waited, but gave up after about an hour (from the stack-of-empty-boxes "curser" I think it was about 30%% done by then. Maybe rendering a few examples on your side an then showing the pre-rendered samples on the web page would work better?

213: "number of nodes in the RNN" - do you mean in the reservoir? Is this the "N" referred to line 123? The first time you mention N in the equations, you could note that you will refer to this number as the size of the network throughout the rest of the paper.
...

218: "loaded in" - I am unclear on what this means exactly.
A high-level talk through would help. First the system is initialised by "loading in" (once we know what that means) M grains that define the space of grains that can be generated. Then the reservoir is manipulated (how exactly - or is loading grains the way they are manipulated) in order to control which attractors are used when the RNN generates it's next grain. Then the reservoir is manipulated again to deliver another grain and so on until we have the number of grains desired to synthesise a sound of a certain length" (or something of that nature).

231: Which parameters are hand-tuned? (I think this becomes clear only later in the paper)

239: "The first 100 audio signal in set a were used to calculate a model" - What are the rest of the samples in the set used for? Or is this because you are representing a single of length 100*mu = 1500?

239: I don't understand the purpose of Figure(3), and line 239 doesn't say anything about it other than that it exists.

Equation 13 for \alpha.sub(scaled) is strange - why not simply say that alpha can be changed over time to control the weight between the current and previous target.
Equation 13 also has an unnecessary layer of parenthesis.

306: "The weight scaling scaleW parameter is a multiplier"
I don't see a scale.sub(w) anywhere - what are you referring to?

336: Are you sure you want to overload the parameter mu (already used for sample length above)?

398: I'm confused how L(washout) and L(train) are expressed in terms of mu, because mu isn't constant for this example. Are L(washout) and L(train) also not constant?

455: A phase shift hardly seems like a problem since it would by inaudible in an A/B comparison between the two signal.

457: "driving audio signals begin to alternate between two frequencies"
Alternate? In time? Clearly the voiced signal is harmonic and made up of multiple frequencies, but I don't understand what you mean by alternating frequencies. Do you mean that your zero-crossing algorithm started alternating between two different signal lengths for segment modelling? Ah - that must be it. I think this could be explained more clearly.

491: the value of scale.sub(alpha) is meaningless without knowing what the original \alpha was - and did \alpha stay inside [0,1] with the changing scale parameter?

Figure 22: The x-axis needs labelling
Figure 22: - The heat map for these spectrograms doesn't correspond very well to what one hears - nor what what one sees in the wave file (S13). A different colour map (of the equal-luminance variety?) may correspond better to perception.

511: "tone"
timbre?

511: variation in Wscaling.
Wscaling in what equation?

526, 548: Scale.sub(w) from what equation?

608-609: generative generation ....


It's great that you provide visual and audio examples to help the reader asses the output.

·

Basic reporting

The paper is well written overall, and has a good structure. The writing is sometimes a bit wordy or convoluted, however, which reduces clarity a bit.

The article provides a good amount of figures and supplementary material.

The research question is not posited very clearly in the article. While it is clear that the overall goal is to construct a sound synthesis system, it is not made clear how it should differ from previously existing approaches, why such a novel system is needed, and which gap in literature is addressed in the process.

Some figures do not flow with the text layout-wise, ending up alone on a page with blank space around them.

While most figures are informative, figure 3 provides no information and can be removed. Some figures are also a bit unfinished, including figure 22, which needs an x-axis labelling, figure 8 having a gray instead of a clear background and no explanation of the meaning of the red dots in the caption, and figure S2 whose values should be rounded to avoid numbers such as 0.7000000001.

For more details, see the general comments section.

Experimental design

The experiments appear novel and interesting, and they give an indication of the potential of the method.

However, I believe they are also far too limited in scope. Please refer to the general comments section for more detail.

Validity of the findings

While the results for the individual sounds used in each experiment appear to be derived properly, the lack of evaluation metrics and the improper scope of the experiments severely limit the possible conclusions. Please refer to the general comments for more details.

Additional comments

Firstly, I would like to mention that I am not familiar enough with reservoir computing and conceptors to fully assess the proposed approach, particularly with respect to the model’s design.

However, my main criticism is related to the evaluation of the approach, and the conclusions that are drawn and can be drawn from the findings.

In general, there are many hyper-parameters (shown in the tables for each experiment), and it is not clear how these were determined. Furthermore, they are set differently for each experiment, and almost every experiment contains of only a single sound. Adapting the hyper-parameters as well as the model parameters all to a single sound each time creates a serious risk of overfitting, which is not addressed in the paper. Also, it appears that parameters need to be adjusted for each experiment as the model is quite unstable, and even further tweaks to how the approach works are often implemented (experiment 2, experiment 3) to make the method work at all. It also seems that the windowing method to create sequence chunks does not work all the time, and needs to be hand-tuned to the given sound. All of this combined raises serious doubts about how generally applicable the method is. This concern is not reflected in the paper. In contrast, it is even stated that their method (“CCRNN”) “can reproduce arbitrary waveforms for granular inspired sound synthesis.” (L462).

Therefore, I would suggest to evaluate the reconstruction ability on a dataset of multiple sounds, without manually optimising parameters to each sound individually, to show generality, or to phrase the claims in the paper in a much less general way.

In terms of results, we further find that there are no experiments to obtain perceptual ratings of any kind for any of the outputs of the model. While an objective NRMSE measure is provided for the reconstruction experiments, for the synthesis/manipulation experiments there is a severe lack of quantiative evaluation. The few quantitative results are also hard to interpret since there are no baseline models being evaluated to compare to.

Generally, the evaluation relies too much on single examples. This applies to experiments 1, 2, 3. Experiment 4 gives the NRMSE for a particular pitch of a generated square wave, but does not plot or average all NRMSE’s for a whole range of pitches. Experiment 5 generates a single sound only. While these exapmles illustrate the working of the method very well, it is not possible to draw sufficiently broad conclusions from them.

For the reconstruction experiments, it should be made clear how much the input sound can be compressed by the model, since perfect reconstruction is trivial if the RNN memory is sufficiently large. This is a serious concern in the current experiments – for example in experiment 1, a 900-node RNN is trained on 100 15-sample signals, along with a conceptor for each segment, suggesting that there could be a very low amount of compression required. Preferably, a compression ratio should be given along with the reconstruction error that shows the relative reduction in the number of bits achieved.

For the experiments involving synthesising new sounds, there is no evaluation besides a subjective interpretation of the author in most cases, including “Extending sound synthesis with conceptor logic”, “Sound morphing with interpolated conceptors”, and experiment 5. It is also stated that pitch shifting can be achieved with the method, but no comparison with other pitch shifting methods is performed. The method is stated to extend granular sound synthesis, but no granular sound synthesis method is used for comparison. While it is repeatedly stated that the significant variety in the model outputs can be interesting for musicians to create new sounds, this subjective claim is not empirically validated in any way. For this set of experiments, some metric(s) of success should be established, such as how interesting the sounds are perceived to be by participants in listening experiments, or the perceived quality of pitch-shifting.

CLARITY ISSUES:

The following are issues with understanding the paper’s content, which probably result from unclear writing. However, in case I am missing something here, this can be discarded.

In the basic model, x_target(n+1) is computed from x(n), and then x(n+1) from x_target(n) and x_target(n+1). With conceptors, this is apparently changed to computing x(n+1) directly from x(n) through the conceptor C (see eq. 10, 11). However, in eq. 13 the interpolation equation originally used in the base model description is seemingly modified, which however is not used since it was replaced by the conceptor transition function. What is behind this apparent mismatch?

Logical operations on conceptor matrices are executed, however it seems that the conceptor matrix arises from an SVD and thus can have real values, and is not binary. How do these logical operations then work?

MINOR REMARKS

L303 How do you precisely limit scale_a?

L292 This chapter seems out of place or at least needs an introduction, since we are moving away from simple reconstruction towards synthesising new sounds – why is that apparently a subsection of experiment 1 still?

L239 Why only the first 100? The whole signal has more than that

Model introduction (“Basic models”) structure could be improved, instead of putting all equations at once upfront, and then explaining all variables in a long block of text, explain each equation step by step and introduce variables along the way

“Loading patterns into the model” chapter is confusing since it is not clear what the actual optimisation objective is, and what is being optimised (is W* and W equal to W as introduced in Eq (1)?) and what parameters are optimised over (only W, or also W_out?).

L195 “Error and similarity” section: It should be made more clear that this is the main evaluation metric used in the experiments, and how it will be used (namely, compare reconstructions to original input)

L201 “Conceptular synthesis” - the first paragraph directly introduces a novel concept, but only justifies it much later. Considering the model description preceding this section, it would be better to have a section explaining why the above proposed model is not directly applicable, and then motivate conceptular synthesis as a solution

Variable names and subscripts are sometimes full words, this detracts from readability and should be shortened

L336 “mu” is chosen as variable for morphing factor although it is already used for the chunk size parameter previously (set to 15 in exp. 1)

L124 “input signal vector a(n) (size 1)” - “size 1” is confusing here, since the vector should have length proportional to the signal duration. Also in L135 suddenly a gets another index j to denote multiple input vectors, which it did not have before!

“sparsely connected with probability of 10.0/N” - what happens when N < 10?
L145 – It is not clear at this point whether time-steps start at 0, or 1

L139-142 states we have to optimize W and W_in so that the given equation (L141) holds, but equations 4 and 5 deal with optimising W and W_out instead – why the discrepancy?

“to field of sound synthesis” → “to the field of sound synthesis”

L236 it might be more understandable to define “a” more precisely by showing its minimum and maximum index. Also it is confusing to call it “set” but use matrix/vector notation where inputs are ordered

L389 “presents” -> “present”

---

## Round 0.2 · Minor Revisions

The reviewers are all happy with the efforts you've made in response to their initial feedback. There are various minor issues that they raise that should be easy to deal with.

In addition, Reviewer 3 makes a good point about reconstruction and model complexity. Their request, either to perform cross-validation or to provide some clear indication of model complexity alongside reconstruction evaluation stats, seems to me well-motivated and I hope you can attend to this.

Reviewer 1 ·

Basic reporting

The writing can still be further improved.

Experimental design

My concerns were addressed.

Validity of the findings

The results are not very strong but my concerns were addressed.

·

Basic reporting

This revision has addresses issues in my previous review.

Experimental design

This revision has addresses issues in my previous review.

Validity of the findings

This revision has addresses issues in my previous review.

Additional comments

Minor points:
There is still an extra nesting of parens in equation 2.

143 Win is generated using linear random values between −1 and 1, and scaled using input scaling factor gamma^ input
- why would you need to scale random numbers that have just been generated to be in [-1,1]?

146 b is a vector of y biases, which are generated from a linear random distribution between −1 and 1,
147 and scaled with bias scaling factor gamma^ bias
- why would you need to scale random numbers that have just been generated to be in [-1,1]?

There is still an extra nesting of parents in equation 14: ((1−a)x(n))

There is still an extra nesting of parents in equation 15.

I think it would be helpful to have a link to audio for each example for which you have an image.

·

Basic reporting

The writing style is clear and professional, and noticeably improved compared to the first submission as many of the reviewer's suggestions have been taken into account.

Some passages are still a bit long-winded and could be improved by shortening them without losing in content, such as the discussion of related work in the beginning of the "Conceptular synthesis" section.

Figures have been much improved, as well as most of the equations.

The related work section remains well written and gives a good overview of recent works.

Minor issues are noted in the "general remarks" section.

Experimental design

The experimental design has been improved substantially after the first round of review and now features a proper baseline method for comparison, where applicable. Hyper-parameters are now more clearly explained and also handled in a more uniform fashion.

For the evaluation, an MFCC error to measure reconstruction fidelity was introduced, which is a good choice as it should result in a more perceptually relevant metric than the previously used RMSE, which was also affected by phase differences.

A major concern in experiments involving the reconstruction of sounds is still the risk of overfitting the models to the input sound. The authors argue, implicitly by making reconstruction error the evaluation metric, that the model should be able to reproduce the input audio exactly, but do not take into account model complexity at the same time.
In that case however, a simple look-up table that saves the whole audio sample would be the optimal "model" with perfect reconstruction error, but not very useful for any other task such as manipulating the audio in a semantically meaningful way, as it has not built a compressed representation of the input that can be manipulated to make changes to the audio.
Therefore, ideally a cross-validation by testing the trained models on separate audio data should be performed, or the model complexity itself should be included as another metric as it should be minimised alongside the reconstruction error, following the Bayesian paradigm of model selection that also factors in the prior probability of each model.
Authors find that CCRNNs with more hidden nodes tend to perform better, which could also be explained by this overfitting phenomenon as the model can more easily memorise the input data.

On the other hand, if the goal is not simply sound reconstruction and generation in the first place, but rather meaningful sound manipulation, ideally evaluation metrics and experimental setups should be changed to measure how well musicians use these model to manipulate sounds.
At the same time, I acknowledge the difficulty in evaluating in such a manner, so I suggest at the minimum to clearly report the complexity of all models used or performing cross-validation as discussed above, along with inserting remarks that discuss overfitting as a potential issue.

Validity of the findings

The findings are well articulated and accurately reported overall, limitations of the method are discussed sufficiently.

One exception to this is their argument that the chaotic and unstable nature of their (and related non-linear) methods is somehow inherently a benefit from a musical perspective as it provides a surprise element. This is not only elaborated in a too long-winded fashion, especially in the paragraph starting in L681, but should be marked more clearly as speculative, or alternatively, references should be given that show a preference of musicians for such novelty when using these tools.

In terms of the validity of the reported results, the main conceptular synthesis experiment has been much improved and therefore the resulting conclusions are quite robust.

However, the conceptillator experiments still rely quite heavily on single examples, which makes the proof of the generality of their application a bit less convincing.
Particularly, the pitched square wave oscillator is only evaluated quantitatively in terms of reconstruction error with a single pitch, and only qualitatively across a pitch range. Here it would be more convincing if reconstruction loss is measured for a range of different pitches, and also the reconstruction loss when adapting a model trained with a particular pitch to output a different one. For example, with a model trained on 65Hz, and required to produce a 110Hz output, record the minimum reconstruction error achieved with the best setting of the leak rate parameter, and repeating this process for a variety of pitch combinations would then give a quantiative measure of how well pitch can be controlled via the leak rate parameter.

Additional comments

MINOR REMARKS

It should be explained what z in Equation (1) is supposed to be below the equation

Equation (2) has no introductory explanation to it

L158 onwards – it is a bit unclear what set of weights is optimised – W* or W („reservoir weights W* are adapted“ then later „W is optimised“). Is W* a separate set of weights from W? It is stated that „the reservoir with weights W* is driven … using equations 1 and 2“, but equations 1 and 2 contain W and not W*.

L158 onwards – there is no mention of how the input weight matrix Win is handled. Is it just randomly initialised at the beginning of optimisation? Is it kept throughout the whole procedure?

L175 „The driving signals from steps from timesteps“ grammatical error, also mention which driving signals by using the mathematical notation, e.g. is that the input a?

Equation 11 is of the form a = b * a. While I understand the intended meaning behind the equation, it would be better to state beforehand that the result of Equation (2) is renamed to x* or similar for the following equation, and then use x* instead of on the RHS.

„Conceptular synthesis“ section – it could be clarified more clearly in the beginning that this is a concept introduced in this paper and makes use of the above introduced methods, as there is a lack of transition from the previous methods section. Also the discussion of related work could be somewhat shortened by transferring it to the related work section where applicable


L283/284 – There should be a bit more explanation on why the presence of high-frequency information can be a problem for the training process apart from just the „distortion“ it. One would naively except the generation system to be able to synthesise waveforms with non-zero starting amplitudes as well?

L312 „was training to resynthesise a kick drum“ → „was trained“

L346 matrix dimensions of X have no proper whitespace

L374 „The use of non-deterministic weight values“ - unclear wording – is this meant to be „non-deterministically chosen initial weight values“?

L387 „models at a higher resolution (ψ = 900)“ does 900 not reflect model size, not (audio) resolution?

Equation (17) is lacking an introductory statement

L396 Results could feature the means/medians of the MFCC errors for both models to have a clearer comparison in addition to the violin plot

Figure 10 caption: „linearly from 0 – 2“ → „linearly from 0 to 2“

Figures 14 and 15 are in the conclusion section when they should appear a section earlier

---

## Round 0.3 · accepted · Accept

Thank you for addressing the reviewers' comments cogently. Congrats on the completed paper.